# LexiCon: a Benchmark for Planning under Temporal Constraints in Natural Language

**Periklis Mantenoglou, Rishi Hazra, Pedro Zuidberg Dos Martires**
Örebro University, Sweden
`{periklis.mantenoglou, rishi.hazra, pedro.zuidberg-dos-martires}@oru.se`

**Luc De Raedt**
Örebro University, Sweden & KU Leuven, Belgium
`luc.deraedt@kuleuven.be`

## Abstract

Owing to their reasoning capabilities, large language models (LLMs) have been evaluated on planning tasks described in natural language. However, LLMs have largely been tested on planning domains without constraints. In order to deploy them in real-world settings where adherence to constraints, in particular safety constraints, is critical, we need to evaluate their performance on constrained planning tasks. We introduce LexiCon—a natural language-based (Lexi) constrained (Con) planning benchmark, consisting of a suite of environments, that can be used to evaluate the planning capabilities of LLMs in a principled fashion. The core idea behind LexiCon is to take existing planning environments and impose temporal constraints on the states. These constrained problems are then translated into natural language and given to an LLM to solve. A key feature of LexiCon is its extensibility. That is, the set of supported environments can be extended with new (unconstrained) environment generators, for which temporal constraints are constructed automatically. This renders LexiCon future-proof: the hardness of the generated planning problems can be increased as the planning capabilities of LLMs improve. Our experiments reveal that the performance of state-of-the-art LLMs, including reasoning models like GPT-5, o3, and R1, deteriorates as the degree of constrainedness of the planning tasks increases.

## 1 Introduction

Planning with constraints is commonly required in problem-solving settings, ranging from resource allocation and scheduling [32] to ensuring safety in reinforcement learning [1, 15, 56, 14]. Several planning specification languages have been proposed [13, 38, 30, 20], including formalisms with constraints [16]. However, specifying the complex, possibly compositional, constraints of an environment in a formal language is rather intricate, as it requires, inter alia, significant domain expertise. Other common ways of integrating constraints in planning problems is via penalties in a reward function [28, 41, 10] or through the physics engine of the environment [7, 46]. These solutions are also challenging for non-experts, while, after tightly integrating constraints into an environment, they are often difficult to alter if needed. We address these limitations by enabling the human user to communicate constraints *directly* to the planning agent, via natural language (NL).

The advent of large language models (LLMs), trained on vast textual corpora, has made NL-based planning increasingly feasible. However, whether LLMs possess the reasoning capabilities required for effective planning remains an open question. Some works argue that LLMs can perform reasoning, and even act as *zero-shot planner* [52, 26, 23], while others critically highlight their limitations [50,

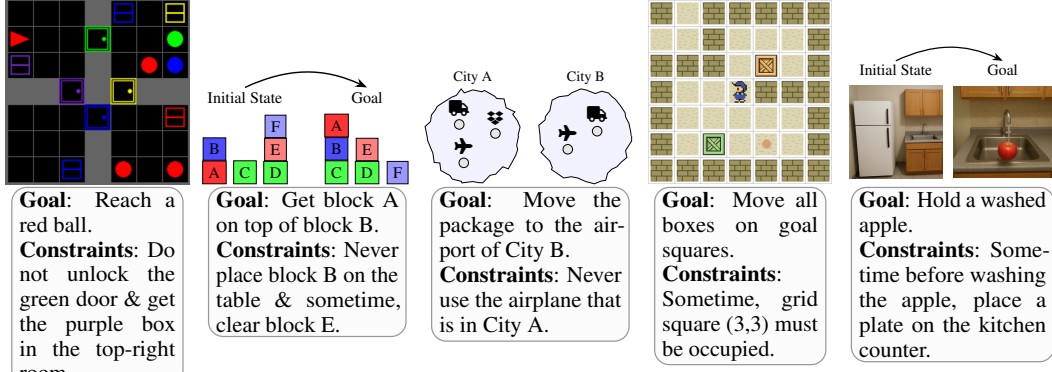

Figure 1: Constrained problems on environments supported in LEXICON. From left to right: BabyAI [6], Blocksworld [19], Logistics [30], Sokoban [12] and AlfWorld [43]. A constrained planning task is specified by an initial state, a goal, and a set of constraints to be respected.

11, 44]. In particular, LLM-based planning methods are often inefficient, lack formal guarantees, and incur high computational costs due to the generation of numerous "thinking tokens" [25, 51].

As LLMs are increasingly deployed in domains such as robotics [24, 29], travel planning [54], tool use [42], scientific discovery [55], and healthcare [45]—all of which demand planning and reasoning under constraints—it becomes crucial to rigorously assess their constrained planning capabilities. To this end, we make the following contributions.

1. **Extensible Benchmark.** We introduce LEXICON, an extensible NL-based benchmark for planning with temporal constraints specified on state-trajectories, which is publicly available[1]. It comprises two core components: a symbolic reasoning engine and a translator, which together enable the following functionalities.

    – **Constrained Problem Generation.** This module takes as input an unconstrained planning problem (described in a formal language) and introduces constraints to it, while making sure that it remains solvable. The reasoning engine generates *task-aware* **constraints** so as to complicate the original problem—resulting in longer solutions compared to its unconstrained version—while guaranteeing that constraints do not subsume one another, which would make them redundant. This leads to a challenging LLM planning benchmark. Crucially, the reasoning engine operates orders of magnitude faster than LLM-based planning, enabling scalable problem generation and evaluation. To interface with LLMs, the translator module converts formal planning problems with constraints into NL, leveraging the compositional structure of these problems to produce NL representations in a systematic manner.

    – **Automated Plan Verification.** The planning capabilities of LLMs are evaluated on the generated NL-representation of constrained planning problems. Subsequently, the reasoning engine automatically verifies whether the LLM-generated plans are correct and/or optimal.

2. **Experimental Evaluation.** We evaluated several state-of-the-art LLMs, including reasoning models like OpenAI o3 [35], DeepSeek R1 [9], Gemini 2.5 Pro [18], Claude 3.7 Sonnet [2], and GPT-5 [34], on benchmarks generated by LEXICON. Using constrained problems of increasing compositional complexity, we found that LLM performance consistently declines with the number of constraints, suggesting that current models do not yet match the performance of formal planning algorithms.

LEXICON supports five environments (Figure 1) and is designed to be extensible. We expect it to remain valuable even as more capable LLMs emerge. As LLMs improve, LEXICON can adapt by generating problems with increased constraint complexity or encorporating new environments, resulting in a flexible, future-proof benchmark that does not rely on static planning problems. Unless LLMs truly acquire algorithmic planning abilities—generalizing across problem instances like symbolic planners—we expect LEXICON to continue serving as an effective tool for assessing their planning capabilities.

---

[1]https://github.com/Periklismant/lexicon_neurips

| Benchmark | Constraints | NL Interface | Automated Curation | Suite Extensibility | Environment Diversity |
|---|---|---|---|---|---|
| BabyAI [6] | ✗ | ✓ | ✓ | ✓ | ✗ |
| AlfWorld [43] | ✗ | ✓ | ✓ | ✗ | ✗ |
| PlanBench [49] | ✗ | ✓ | ✓ | ✓ | ✓ |
| ACPBench [27] | ✗ | ✓ | ✓ | ✓ | ✓ |
| BALROG [37] | ✗ | ✓ | ✓ | ✓ | ✓ |
| Safety Gym [41] | ✓ | ✗ | ✓ | ✗ | ✗ |
| TravelPlanner [54] | ✓ | ✓ | ✗ | ✗ | ✗ |
| Natural Plan [57] | ✓ | ✓ | ✗ | ✗ | ✓ |
| **LEXICON** | ✓ | ✓ | ✓ | ✓ | ✓ |

Table 1: Comparison of simulation benchmarks. "Automated curation" indicates the ability to automatically generate new planning problem instances and verify solutions for those instances. "Suite Extensibility" requires that new planning domains can be added to the benchmark without rewriting its code. "Environment Diversity" indicates that the benchmark supports more than one type of planning domain (e.g., it is not restricted solely to 2D gridworld problems).

## 2 Related Work

Table 1 compares LEXICON with state-of-the-art planning benchmarks. Benchmarking the planning capabilities of LLMs requires an NL interface, which limits the applicability of traditional constrained environments such as Safety Gym [41] that lack NL support. While simulators such as BabyAI [6], gComm [21], and AlfWorld [43] support NL interaction, they do not model constraints and are limited to narrow domains (e.g., 2D grids or household settings). Constrained planning benchmarks like NaturalPlan [57] and TravelPlanner [54] also support NL, but their tasks are either manually curated or carefully constructed offline, resulting in limited extensibility. Additionally, verifying LLM-generated plans in these settings typically requires exhaustively enumerating all valid solutions, which is prohibitively expensive. In contrast, LEXICON supports the generation of a potentially unbounded number of constrained tasks and can automatically verify agent outputs using its reasoning engine. This enables rigorous, scalable evaluation without needing exhaustive (manual) plan enumeration.

While one might consider augmenting planning benchmarks such as PlanBench [49] or BALROG [37] with constraint-handling functionalities, these systems lack the infrastructure to synthesize, solve, and validate constrained tasks in an integrated manner. In contrast, LEXICON was built from the ground up to support automated constraint generation, enforcement, and verification. As LLMs continue to improve in their reasoning capabilities [22], LEXICON provides a principled platform for evaluating them on increasingly complex planning tasks with compositional constraints. Moreover, its reasoning engine is domain-agnostic, facilitating seamless extension to new environments (i.e., suite extensibility). In what follows, we illustrate the planning formalism in LEXICON, and describe its architecture.

## 3 The LEXICON Simulator

### 3.1 Planning Specification Language

LEXICON supports planning problems expressed in PDDL3.0, an extension of the PDDL formal planning language that includes constraints [16].

> **Example 3.1: Constrained Planning in BabyAI**
>
> BabyAI contains problems where an agent needs to navigate the rooms of 2D gridworld, while interacting with objects, to complete some task [6]. Figure 2 (left) shows the initial state of a problem from BabyAI. BabyAI problems are grounded in PDDL; a domain file specifies the object types, the (time-varying) state atoms, and the actions of the domain, while a problem file denotes the objects of the puzzle, the initial state, the goal, and the constraints. In this case, the domain file defines atom `locked(d)`, expressing that door `d`

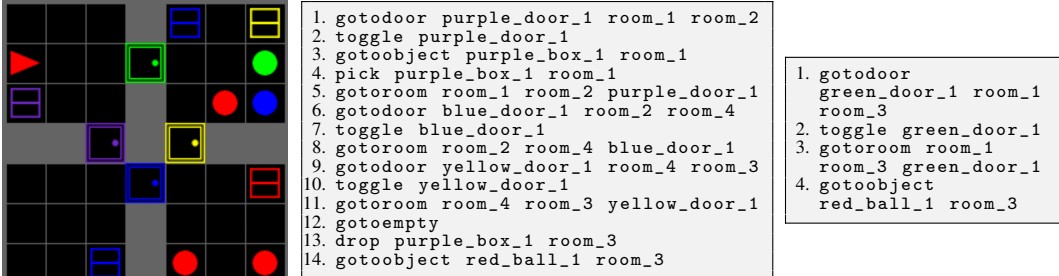

```
1. gotodoor purple_door_1 room_1 room_2
2. toggle purple_door_1
3. gotoobject purple_box_1 room_1
4. pick purple_box_1 room_1
5. gotoroom room_1 room_2 purple_door_1
6. gotodoor blue_door_1 room_2 room_4
7. toggle blue_door_1
8. gotoroom room_2 room_4 blue_door_1
9. gotodoor yellow_door_1 room_4 room_3
10. toggle yellow_door_1
11. gotoroom room_4 room_3 yellow_door_1
12. gotoempty
13. drop purple_box_1 room_3
14. gotoobject red_ball_1 room_3
```

```
1. gotodoor
   green_door_1 room_1
   room_3
2. toggle green_door_1
3. gotoroom room_1
   room_3 green_door_1
4. gotoobject
   red_ball_1 room_3
```

```
1. (:goal (and (exists (?v - ball) (and (objectcolor ?v red) (at ?v)))))
2. (:constraints (always (locked green_door_1))
3.                (sometime (agentinroom room_1))
4.                (sometime-after (agentinroom room_1) (objectinroom purple_box_1 room_3)))
```

Figure 2: Left: The initial state of the constrained planning problem in Example 3.1. The red triangle represents the agent. Bottom: The goal and the constraints of the problem in PDDL3.0. Middle: Optimal plan for this problem. Right: Optimal plan for the corresponding unconstrained problem.

is locked, and action `pick`, outlining the conditions for and the effects of picking up and holding an object. Figure 2 (bottom) outlines the goal and the constraints of the problem. The goal is to reach a red ball, while the constraints dictate that (i) the agent must never unlock the green door, (ii) at some point, the agent must visit room 1 (top-left room), and (iii) some time after visiting room 1, purple box 1 needs to be in room 3 (top-right room).

In LEXICON, we are interested in **optimal planning**, i.e., finding a plan that (1) reaches the goal while satisfying all constraints and (2) has minimum length. Optimal planning on the problem described in Example 3.1 is easy if the constraints are ignored—an optimal plan for the unconstrained problem consists of 4 actions (see Figure 2 (right)). However, the constrained version is significantly more challenging. For example, to satisfy the constraint that the green door must always remain locked, the agent must take a longer path through the purple and blue doors to reach the room containing the red ball—resulting in 14 actions (cf., Figure 2 (middle)).

## 3.2 The LEXICON Architecture

Figure 3(left) illustrates the architecture of LEXICON. The modules between the "Sampler" and the "Translator" implement the constrained problem generator functionality of LEXICON, while the "Verifier" module realises the automated plan verification functionality. We first outline constrained problem generation in PDDL, then its translation into natural language, and lastly our plan verifier.

**Constrained Planning Problem Generator.** We developed a constrained PDDL problem generator, extending the literature with a **task-aware** method for producing constraints for arbitrary PDDL problems. Its task is to generate constrained planning problems along with their optimal cost. The generator first samples an unconstrained problem using a domain file and a (unconstrained) state-goal pair generator, and then computes an optimal plan for the problem using the state-of-the-art planner SymK [47]. This plan, along with the unconstrained problem, is passed to LEXICON's constraint generator, which synthesizes task-aware constraints that (1) preserve feasibility, i.e., the problem still has a solution, and (2) increase the optimal cost relative to the unconstrained version.

### Example 3.2: Constraint Generation in BabyAI

Consider the unconstrained plan in Figure 2. Using LEXICON, we can automatically construct an Always($\phi$) constraint by analyzing the state transitions induced by this plan. The system samples domain atoms and evaluates their suitability for inclusion in $\phi$ based on problem complication, consistency, and non-redundancy. For example, atom `at(red_ball_2)` is excluded since it does not hold in the initial state and thus cannot "always" hold. Similarly, `objectInRoom(red_ball_1, room_3)` is not selected as it holds in all states of the unconstrained plan, thus offering no added difficulty. In contrast, `locked(green_door_1)`

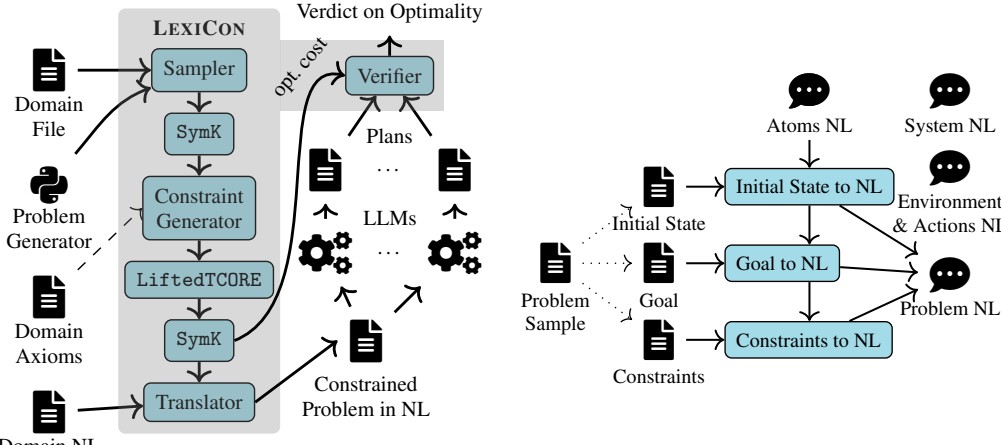

Figure 3: Left: The architecture of LEXICON. Solid arrows denote input/output data transfers. Dashed arrows denote optional input. Right: The translator of LEXICON. Dotted arrows express content extraction.

> is included in $\phi$, as enforcing it prevents use of the green door—forcing a detour through the purple and blue doors—which increases the plan's optimal cost (see Figure 3 (left)).

In LEXICON, users can optionally provide atemporal **domain axioms** to guide the constraint generator toward meaningful, non-conflicting constraints. For example, given the axiom $\forall d: \neg(\texttt{locked}(d) \land \texttt{unlocked}(d))$ and an existing constraint Always($\texttt{locked}(\texttt{green\_door\_1})$), the generator avoids sampling Sometime($\texttt{unlocked}(\texttt{green\_door\_1})$) since it would be unsatisfiable.

Next, we compute the optimal cost of the constrained problem generated by LEXICON, which is necessary to evaluate LLM outputs against ground-truth optimal plans. However, no existing planner supports constrained planning problems with actions that have conditional effects, which are often essential to specify certain domains, such as BabyAI. To overcome this, we compile the constraints away [53, 39, 5], producing an equivalent problem without constraints, which can be solved by SymK. LEXICON uses the TCORE compiler [4] for this translation. To avoid the cost of grounding, we apply a lifted variant of TCORE. Solving the compiled problem with SymK yields a formally verified optimal cost for the original constrained planning problem.

Our constrained problem generator is **compositional**, allowing users to control the number and complexity of constraints, enabling the generation of increasingly challenging benchmarks for future LLMs. It is also **extensible**: to support a new domain, users need only provide a PDDL domain file and an automated initial state–goal generator, avoiding manual problem construction. Domains can also be specified in Python via the Unified Planning framework [31], easing use for non-experts.

**PDDL to Natural Language Translator.** To evaluate LLMs on constrained problems, we first translate them into natural language (NL). As shown in Figure 3 (right), our translator extracts the instance-specific elements—initial state, goal, and constraints—and composes a problem prompt in NL. Since these instance-specific elements are built compositionally from domain atoms, their NL descriptions are generated by combining predefined NL templates for each atom. Domain-level descriptions (e.g., environment and action semantics) are carefully handcrafted per domain.

---

**Example 3.3: NL Translation for BabyAI Problem**

Consider the constrained planning problem in Example 3.1. Figure 4 (top) shows a fragment of the NL description generated for this problem by our translator. Lines 2–5 describe the initial state by listing NL descriptions of atoms that hold initially. The goal—"*reach a red ball*"—is represented by the logical formula $\exists v: \texttt{typeof}(v, \texttt{ball}) \land \texttt{objectColor}(v, \texttt{red}) \land \texttt{at}(v)$, which we translate recursively into NL (lines 7–8). This involves mapping the quantifier to "*There is a ball v such that*", followed by "*The following*

---

```
1. The original state of the world is:
2. 'you are in room_1'
3. 'purple_box_1 is in room_1'
4. 'blue_box_1 is in room_2'
5. <Description of the remaining atoms that hold initially>
6.
7. The task is to bring about the following situation:
8. 'There is a ball v such that 'The following conditions are all true: 'v is red', 'you
   are in front of v'''
9.
10. A valid plan for the abovementioned problem must abide by the following constraints:
11. 'The following expression must hold in every state: 'green_door_1 is locked''
12. 'The following expression must hold in at least one state: 'you are in room_1''
13. 'If expression 'you are in room_1' holds in some state s, then expression '
    purple_box_1 is in room_3' must hold at s or at some state after s'
```

```
1. Provided a planning problem, consisting of an initial state of the world, a final goal
   and a set of constraints, your task is to provide a valid sequence of actions that
   solves the planning problem, i.e., bringing about the goal of the problem while
   satisfying all constraints.
2. You need to provide an optimal plan, i.e., a valid plan whose length is equal or less
   than the length of any other valid plan.
```

Figure 4: Top: Fragment of our natural language description of the constrained problem of Example 3.1. Bottom: System role prompt.

*conditions are all true*", and then enumerating atom-level descriptions. Constraints are translated similarly using this recursive procedure (see lines 10–13).

Our translator is also **extensible**: to support a new planning domain, one only needs to provide (i) an NL description of the environment and actions, and (ii) NL descriptions for each atom. This eliminates the need for instance-specific NL annotations, allowing the translator to operate directly on any generated constrained problem within the domain.

**Automated LLM Plan Verifier.** With LEXICON's modules for generating constrained planning problems in NL in place, we now evaluate LLMs on these problems. Each LLM is given the NL description of a problem along with a fixed system role prompt (Figure 4 (bottom)), instructing it to act as an optimal planner. This prompt is used consistently across all domains.

To assess LLM outputs, LEXICON includes a verifier module (Figure 3 (left)) with three steps: (1) LLM-generated plans are mapped to PDDL actions using the prescribed output format; deviations are corrected by matching the LLM action to the closest domain action, according to the edit distance [33]. (2) The plan is validated using an automated plan validator on the compiled version of the constrained problem produced by LiftedTCORE, leveraging the guarantee that a plan valid for the compiled problem also satisfies the original constrained problem [4]. (3) If valid, the plan is checked for optimality by comparing its length to the optimal cost, which was computed at the problem generation phase.

A rigorous formulation of the constrained planning problem (i.e., with temporal constraints) along with how constrained plans are generated and verified through our reasoning engine is provided in Appendix A.

Figure 5 displays LLM-generated plans for the constrained problem in Example 3.1. The plan on the left was generated by o3; this plan is invalid because it violates the preconditions of the pick action, i.e., the agent attempts to pick up a purple box at a time when it is not facing that box (cf. line 1 of Figure 5 (left) and the starting state in Figure 2 (left)). The plan in the middle was suggested by Claude 3.7 Sonnet with extended thinking. This plan is invalid because the agent attempts to drop an object at a time when it is facing a door instead of an empty position, as required by the preconditions of the drop action (cf. lines 6 and 7 in Figure 5 (middle)). This type of error may be due to LLM state hallucination or loss of state tracking. The plan on the right was produced by R1. This plan ignores the constraint stipulating that the purple box must be placed in the top-right room, and is thus invalid. Next, we present a thorough evaluation of LLMs on benchmarks generated by LEXICON.

```
1. pick purple_box_1 room_1        1. gotoobject purple_box_1       1. gotodoor purple_door_1
2. gotodoor purple_door_1             room_1                           room_1 room_2
   room_1 room_2                   2. pick purple_box_1 room_1      2. toggle purple_door_1
3. toggle purple_door_1            3. gotodoor purple_door_1        3. gotoroom room_1 room_2
4. gotoroom room_1 room_2             room_1 room_2                    purple_door_1
   purple_door_1                   4. toggle purple_door_1          4. gotodoor blue_door_1
5. gotodoor blue_door_1            5. gotoroom room_1 room_2           room_2 room_4
   room_2 room_4                      purple_door_1                 5. toggle blue_door_1
6. toggle blue_door_1              6. gotodoor blue_door_1          6. gotoroom room_2 room_4
7. gotoroom room_2 room_4             room_2 room_4                    blue_door_1
   blue_door_1                     7. drop purple_box_1 room_2      7. gotoobject red_ball_2
8. gotodoor yellow_door_1          8. toggle blue_door_1               room_4
   room_4 room_3                   9. gotoroom room_2 room_4
9. toggle yellow_door_1               blue_door_1
10. gotoroom room_4 room_3         10. gotoobject red_ball_2
    yellow_door_1                      room_4
11. drop purple_box_1 room_3
12. gotoobject red_ball_1
    room_3
```

Figure 5: Invalid plans suggested by LLMs for the constrained problem in Example 3.1.

# 4 LLM Evaluation on LEXICON

## 4.1 Evaluation Setup

Figure 1 displays the domains supported in LEXICON, which are:

- **BabyAI** [6]: an environment with minigrid problems, like our running example.
- **Blocksworld**: a puzzle where an agent rearranges blocks into a target configuration. Constraints may forbid placing certain blocks on the table or require specific sequences of block manipulations.
- **Logistics**: a world consisting of several locations, possibly including packages, trucks and airplanes, where the task is to move all packages to their designated destinations. Constraints may, e.g., forbid the usage of a specific truck or an airport.
- **Sokoban**: a gridworld where an agent has to move a collection of boxes onto target locations. Constraints may indicate, e.g., that a grid square must be occupied or cleared.
- **AlfWorld**: an environment for executing household task, like putting a book in a drawer, washing and slicing an apple, or turning on a lamp. Constraints may, e.g., prohibit the use of certain utensils, or impose a (partial) ordering among sub-tasks.

LLMs were tasked with optimal planning on constrained problems generated by LEXICON. Our experiments ran on a standard PC (Ubuntu 22, Ryzen 7 5700U, 16GB RAM), using each LLM's official API and the maximum allowed token limits for completions and reasoning. The LLM execution parameters for all our experiments are provided in Appendix C.

## 4.2 Evaluation Results

We evaluated 5 LLMs with thinking token generation capabilities, i.e., DeepSeek R1 [9], OpenAI o3 [35], Gemini-2.5 Pro [18], Claude 3.7 Sonnet (Extended Thinking) [2], and GPT-5 [34]. We also tested 4 LLMs that that do not support thinking capabilities, i.e., GPT 4.1 [36], DeepSeek V3 [8], Claude 3.7 Sonnet (no extended thinking), and Gemini 2.0 Pro [17], on benchmarks generated by LEXICON. Each environment consisted of 150 problems, with the number of constraints in $\{1, 3, 5, 7, 10\}$.

**Can LLMs perform constrained planning?** Figure 6 displays our results. All data points were produced over 30 executions. For the sake of comparison, we also included performance measurements on unconstrained problems. **LLMs without explicit thinking typically failed to produce an optimal—or even valid—plan for problems involving more than one constraint**. For this class of models, we only report the best-performing configuration, which was GPT-4.1 with Chain-of-Thought (CoT) prompting [52]. For the performance of the remaining models, see Appendix B. In contrast, reasoning models frequently succeeded in producing optimal plans for problems with a few constraints. However, **the capabilities of reasoning LLMs deteriorated sharply as the number of constraints increased**. In the majority of problems including 10 constraints, most LLMs failed to even produce a suboptimal plan. In the case of o3, e.g., in Blockworld, the optimal planning accuracy over constraint range $\{1, 3, 5, 7, 10\}$ was [76%, 30%, 26%, 10%, 0].

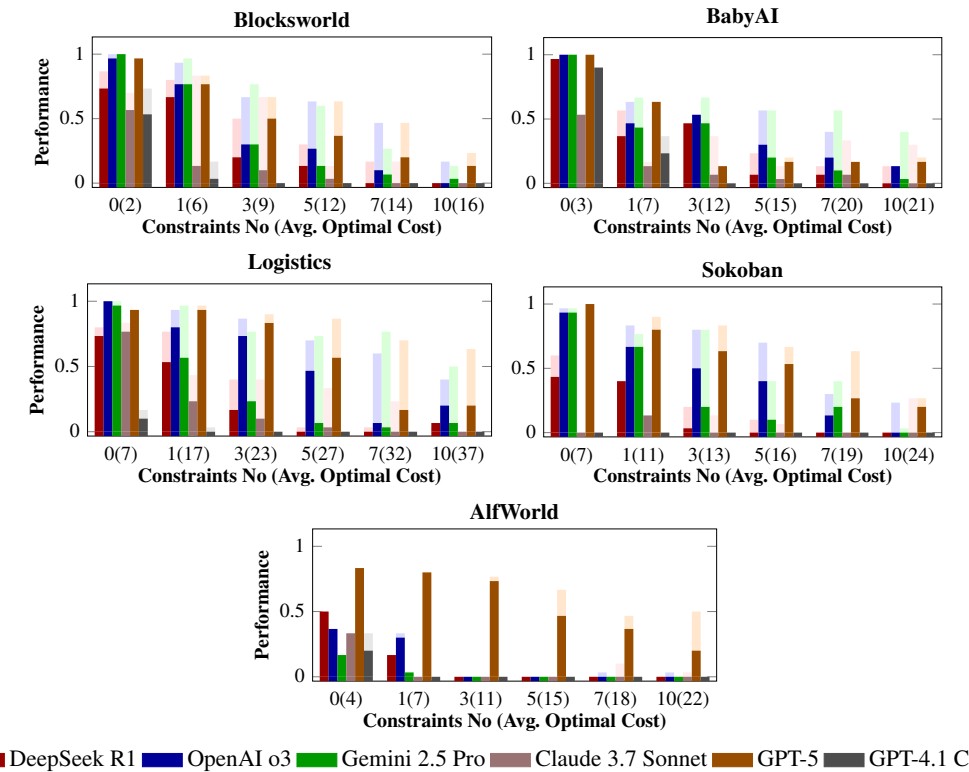

Figure 6: **Performance vs. number of constraints (average optimal cost)**. Performance denotes the percentage of problems solved with an optimal plan (colored bars) or with a valid, but possibly suboptimal plan (background faded-out bars).

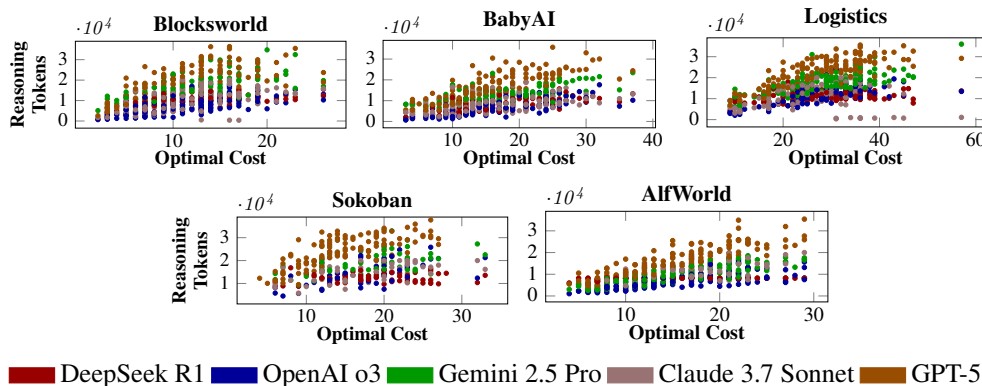

Figure 7: **Correlation between Reasoning Tokens and Optimal Cost**. The Pearson correlation coefficients are as follows. **Blocksworld:** [0.48, 0.6, 0.67, 0.66, 0.63]; **BabyAI:** [0.52, 0.68, 0.7, 0.74, 0.8]; **Logistics:** [0.16, 0.37, 0.73, 0.71, 0.77]; **Sokoban:** [0.53, 0.12, 0.57, 0.62, 0.72]; **AlfWorld:** [0.46, 0.74, 0.86, 0.55, 0.87].

Interestingly, as shown in Figure 7, we observe that the **number of thinking tokens increases with the optimal plan length** (i.e., optimal cost), suggesting that the reasoning models engage in deeper reasoning when the task demands it. However, as highlighted in Figure 5, their *soundness* declines for longer plans, due to: invalid actions from precondition violations (e.g., attempting to pick up a box without facing it), hallucinated states (e.g., perceiving an empty space where a door exists), misinterpreted constraints, and loss of state tracking.

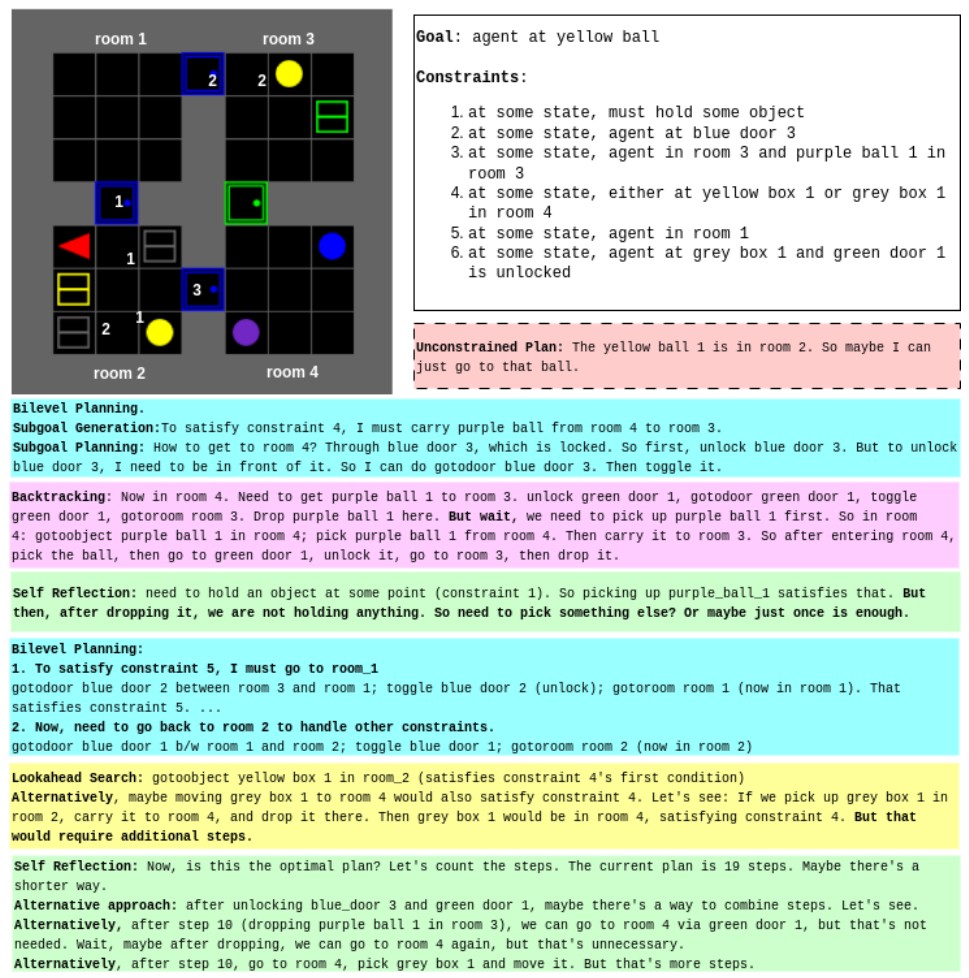

Figure 8: **Planning Traces from R1**. The white box displays the goal and constraints. On the left is the initial observation from the BabyAI environment. Colored boxes indicate model behaviours: cyan for bilevel planning, green for self-reflection, yellow for lookahead search, and violet backtracking. We also show the 1-step unconstrained plan generated by R1 for the same goal in salmon - - -.

**Do LLMs show structured planning behaviour?** We qualitatively analysed the reasoning traces of R1 by annotating its "thinking" steps and mapping them to classical planning strategies. (The reasoning traces of the LLMs that performed better on our benchmark were not available to us.) Figure 8 shows a representative example. We observed the following behaviours:

- **Bilevel Planning**. R1 decomposes the goal into high-level subgoals and performs subgoal planning.
- **Backtracking**. R1 can backtrack and generate a more optimized subgoal plan (e.g, instead of having to go back to pick an object, carry it with you).
- **Lookahead Search**. R1 generates multiple rollout paths and selects the optimal action.
- **Self-Reflection**. R1 frequently re-evaluates the state and the selected actions, checking constraint satisfaction and exploring alternatives, towards optimising the plan.

Despite these interesting behaviours, **R1 does not show the structured search behaviour necessary for optimal planning**, like maintaining different search paths simultaneously. Instead, it tries to generate a valid plan that satisfies all constraints, and subsequently attempts to shorten that plan, towards finding an optimal one; this is not guaranteed to work for arbitrary planning problems.

**Can LEXICON enable real-time evaluation of LLM Planners?** A key advantage of LEXICON is its ability to generate arbitrary constrained planning problems on demand. This allows for on-the-fly evaluation of LLMs on problems of varying complexity, rather than relying on static, offline

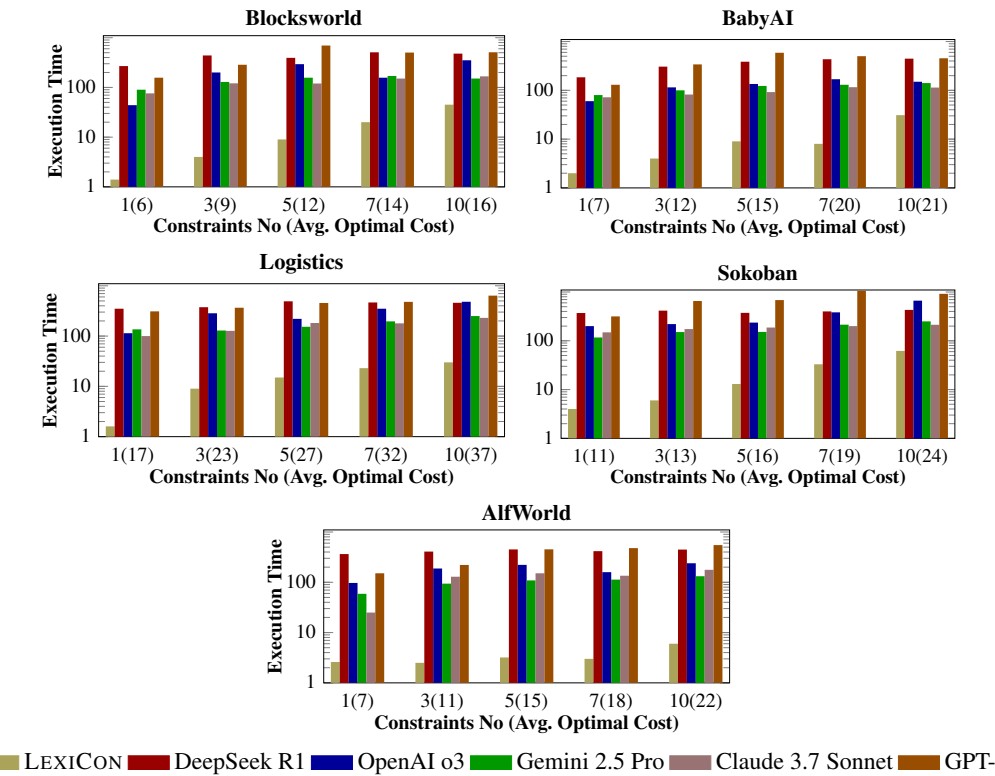

Figure 9: **Execution Time vs. number of constraints (average optimal cost)**. The vertical axes show execution time in seconds. Standard deviations are small, and thus omitted.

benchmarks. In our setup, LEXICON generates a new problem while the LLM is still solving the previous one, enabling a seamless and adaptive evaluation pipeline. To test this, we compared the average time LEXICON takes to generate and verify a problem (in natural language and PDDL) with the average time an LLM takes to solve it[2]. As shown in Figure 9, **LEXICON is roughly one order of magnitude faster than LLM-based planning, making real-time evaluation feasible**.

## 5 Summary & Future Work

We proposed LEXICON, an extensible NL-based benchmark generator for planning under temporal constraints. Our generator is able to produce task-aware constraints for an arbitrary planning problem and verify solutions suggested by LLMs at scale. Our experiments showed that there is a limit of problem constrainedness that LLMs cannot cope with, even for models with reasoning capabilities. We aim to extend LEXICON with partially-observable environments and uncertain observations, as well as a wider class of constraints, including constraints on actions and on continuous states, paving the way for evaluating language-based agents on real reinforcement learning settings.

## 6 Limitations

Our simulator does not support parallel episode execution, unlike standard RL environments such as MuJoCo [46] or Atari [3], which can be parallelized using tools like AsyncVectorEnv (Gymnasium) [48] or SubprocVecEnv (Stable-Baselines3) [40]. In our case, multiprocessing is fully utilized for backend tasks such as generating feasible episodes. Furthermore, episode generation is significantly slower ($[1, 100]$ s) due to the complexity of constraint satisfaction and simulation, limiting scalability compared to environments that support fast, parallel rollouts.

---

[2]Note that LLM solve time also depends on API latency, though LEXICON remains significantly faster.

## Acknowledgments and Disclosure of Funding

This work was supported by the Wallenberg AI, Autonomous Systems and Software Program (WASP) funded by the Knut and Alice Wallenberg Foundation. This work was also supported by the Research Foundation - Flanders (FWO) under contract no G097720N.

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

# Appendix

This document contains supplementary material for our paper, along with code execution and reproducibility instructions for our experiments. Its structure is the following. Appendix A describes LEXICON's reasoning engine. Appendix B exemplifies indicative errors in LLM planning, while also highlighting some experimental results that were omitted from the paper due to space limitations. Appendix C provides the hyperparameters set for each LLM in our experiments, along with the steps for running our code and reproducing our experiments.

## A  Reasoning Engine

First, we specify the class of planning problems that may be generated and have candidate solutions verified by LEXICON's reasoning engine. Subsequently, we describe its two main modules: the constraint planning problem generator and the automated LLM plan verifier.

### A.1  Class of Planning Problems in LEXICON

LEXICON may generate and verify candidate solutions for planning domains expressed in a PDDL fragment that includes the following syntactic components.

- Basic STRIPS, i.e., actions with conjunctive preconditions, and atom addition and deletion effects [13].

- ADL, i.e., equalities, actions with negated, disjunctive and quantified preconditions, as well as conditional and universally quantified effects [38].

- The qualitative state-trajectory constraints found in PDDL3.0 [16].

We formulate this fragment of PDDL, loosely following [4] for the notation, and using the term "constraint" to refer to a qualitative state-trajectory constraint of PDDL3.0 for brevity.

A constrained planning problem is a tuple $\Pi^C = (F, A, I, G, C)$, where $F$ is a set of atoms, $A$ is a set of actions, $I \subseteq F$ is an initial state, $G$ is a formula over $F$ denoting the goal of the problem, and $C$ is a set of constraints. Each action $a \in A$ comprises a precondition $Pre(a)$, which is a formula over $F$, and a set of conditional effects $Eff(a)$. Each conditional effect in $Eff(a)$ is an expression $c \triangleright e$, where $c$ is a formula and $e$ is a set of literals, both constructed based on the atoms in $F$. We use $e^+$ (resp. $e^-$) to denote the positive (negative) literals in $e$. A state $s \subseteq F$ contains the atoms that are true in $s$. An action $a$ is applicable in state $s$ if $s \models Pre(a)$, and its application yields state $s' = (s \setminus \bigcup_{c \triangleright e \in Eff(a) : s \models c} e^-) \cup \bigcup_{c \triangleright e \in Eff(a) : s \models c} e^+$, which we often denote with $s' = s[a]$.

LEXICON supports the following types of constraints: Always, Sometime, AtMostOnce, SometimeBefore and SometimeAfter. Considering grounded formulas $\phi$ and $\psi$ over $F$ in negation normal form, and a sequence of states $\sigma$ over $F$, these constraint types are defined as follows:

- $\sigma \models \mathsf{Always}(\phi)$ (or $\mathsf{A}(\phi)$) iff $\forall s \in \sigma$: $s \models \phi$.

- $\sigma \models \mathsf{Sometime}(\phi)$ ($\mathsf{S}(\phi)$) iff $\exists s \in \sigma$: $s \models \phi$.

- $\sigma \models \mathsf{AtMostOnce}(\phi)$ ($\mathsf{AO}(\phi)$) iff $\phi$ is true in at most one continuous subsequence of $\sigma$.

- $\sigma \models \mathsf{SometimeBefore}(\phi, \psi)$ ($\mathsf{SB}(\phi, \psi)$) requires that, if $\exists s \in \sigma : s \models \phi$, then there is a state $s'$ before $s$ in $\sigma$, such that $s' \models \psi$.

- $\sigma \models \mathsf{SometimeAfter}(\phi, \psi)$ ($\mathsf{SA}(\phi, \psi)$) requires that, if $\exists s \in \sigma : s \models \phi$, then $s \models \psi$ or there is a state $s'$ after $s$ in $\sigma$ such that $s' \models \psi$.

Given a constrained planning problem $\Pi^C = (F, A, I, G, C)$, a plan $\pi$ for $\Pi^C$ is a sequence of actions $(a_0, \ldots, a_{n-1})$ from set $A$. $\pi$ is a valid plan for $\Pi^C$ iff there exists a sequence of states $\sigma = (s_0, \ldots, s_n)$ such that $s_0 = I$, $\forall i \in \{0, \ldots, n-1\}$ we have $s_i \models Pre(a_i)$ and $s_{i+1} = s_i[a_i]$, $s_n \models G$, and $\forall q \in C$ we have $\sigma \models q$. We define the cost of a plan as the number of actions it includes. An optimal plan $\pi^*$ for a problem $\Pi^C$ is a valid plan whose cost is minimal among all valid plans for $\Pi^C$, i.e., there is no valid plan for $\Pi^C$ that has a lower cost than $\pi^*$.

## A.2 Constrained Planning Problem Generator

We focus on the "Constraint Generator" of LEXICON, as the remaining modules of our reasoning engine that are used for problem generation are off-the-shelf planners and compilers (see Figure 3). Our constraint generator receives as input an unconstrained PDDL problem $\Pi = (F, A, I, G)$ and an optimal plan $\pi^*$ for $\Pi$, and outputs a constrained PDDL problem $\Pi^C = (F, A, I, G, C)$. The challenge here is to construct constraint set $C$ in an informed manner, considering problem $\Pi$ and plan $\pi^*$. In particular, we may add a constraint $q$ in $C$ only if $q$ is a meaningful constraint given $\Pi$ and $\pi^*$, i.e., the inclusion of $q$ makes $\pi^*$ an invalid plan for $\Pi$, potentially complicating the planning problem, while maintaining problem solvability and being non-redundant with respect to the constraints that were previously added in $C$.

In order to produce such a meaningful constraint $q$, we proceed as follows.

1. We identify a set of conditions under which a literal is not suitable for inclusion in $q$, in the sense that its inclusion potentially results in $q$ not being meaningful for the problem.

2. We sample literals that do not satisfy the conditions identified in the previous step, and consider whether they should be included in $q$. For each sampled literal $l$, we verify that it is consistent with, and not subsumed by, the literals that were previously added in $q$, taking into account a (possibly empty) set of domain axioms. If this is the case, then we add $l$ in $q$. We continue this process until $q$ has reached a specified degree of compositionality, which may be controlled by the user.

3. We verify that the generated constraint $q$ is consistent with, and not subsumed by, the constraints that were previously added in $C$, in which case we add $q$ in $C$.

We continue this process until the size of $C$ has reached the number of constraints requested by the user.

---

**Algorithm 1** Always Constraint Generator

---

**Require:** State changes $\sigma$ induced by executing plan $\pi^*$, unconstrained problem $\Pi = (F, A, I, G)$, constraint set so far $C$, domain axioms $D$, possible user parameter values $cfg$
**Ensure:** New constraint set $C \cup \{q\}$
1: $op, l\_no \leftarrow sample\_parameters(cfg)$, $literals \leftarrow \emptyset$
2: **for** $l\_no$ iterations **do**
3:     $l \leftarrow sample\_literal(F)$
4:     **if** $l \rightarrow G$ **or** $G \rightarrow \neg l$ **or not** $(I \models l)$ **or** $\forall s \in \sigma : s \models l$ **then goto** 3
5:     **for** $l'$ **in** $literals$ **do**
6:         **if** $l = l'$ **or** $(op = \wedge$ **and** $D \models \neg(l \wedge l'))$ **then goto** 3
7:     $literals.\text{append}(l)$
8: **if** $op = \wedge$ **then** $\phi \leftarrow \bigwedge_{l \in literals} l$ **else** $\phi \leftarrow \bigvee_{l \in literals} l$
9: $q \leftarrow \mathsf{A}(\phi)$
10: **for** $q' \in C$ **do**
11:     **if** $q' = \mathsf{A}(\phi')$ **and** $(D \models (\phi \rightarrow \phi')$ **or** $D \models (\phi' \rightarrow \phi)$ **or** $D \models \neg(\phi \wedge \phi'))$ **then goto** 1
12:     **else if** $q' = \mathsf{S}(\phi')$ **and** $(D \models (\phi \rightarrow \phi')$ **or** $D \models \neg(\phi \wedge \phi'))$ **then goto** 1
13:     **else if** $q' = \mathsf{AO}(\phi')$ **and** $(D \models (\phi \rightarrow \phi')$ **or** $D \models \neg(\phi \wedge \phi'))$ **then goto** 1
14:     **else if** $q' = \mathsf{SB}(\phi', \psi')$ **and** $(D \models (\phi \rightarrow \phi')$ **or** $D \models (\phi \rightarrow \psi')$
15:                                  **or** $D \models \neg(\phi \wedge \phi')$ **or** $D \models \neg(\phi \wedge \psi'))$ **then goto** 1
16:     **else if** $q' = \mathsf{SA}(\phi', \psi')$ **and** $(D \models (\phi \rightarrow \psi')$ **or** $D \models \neg(\phi \wedge \phi')$ **or** $D \models \neg(\phi \wedge \psi'))$ **then**
17:         **goto** 1
18: **return** $C \cup \{q\}$

---

The above procedure for generating a constraint is adapted for each possible type of constraint. Constraint consistency and subsumption, e.g., is defined differently for each constraint type. As an example, Algorithm 1 outlines the procedure for constructing an Always constraint $\mathsf{A}(\phi)$. We start by sampling a Boolean operation $op$ and a number of literals $l\_no$ for $\phi$, taking into account the parameters that are optionally provided by the user (see line 1 of Algorihtm 1). Subsequently, we generate $l\_no$ literals that are suitable for inclusion in $\phi$ (lines 2–7). We sample a literal $l$ based on the atoms of the problem $F$ (line 3), and then evaluate a set of conditions such that, if $l$ satisfies one

of them, then including $l$ in $\phi$ is not meaningful with respect to constraint $\mathsf{A}(\phi)$. For instance, it is not meaningful to include $l$ in $\mathsf{A}(\phi)$ if (i) $l$ implies the goal $G$, as, in non-trivial problems where $I \not\models G$, $l \to G$ implies that $l$ does not hold in the initial state $I$, and thus cannot "always" hold; (ii) $G$ implies $\neg l$, because then $l$ cannot hold in the final state of a plan that brings about $G$; (iii) if $l$ does not hold in $I$; or (iv) if $l$ is satisfied in every state in the sequence $\sigma$ induced by executing plan $\pi^*$, as, in that case, $\mathsf{A}(l)$ is satisfied by optimal plan $\pi^*$ of the unconstrained problem, and thus adding $l$ in $\mathsf{A}(\phi)$ may not lead to a more complicated problem. If any of the above conditions holds, then we drop $l$ and sample another literal for our constraint (line 4). Additionally, we resort to resampling if $l$ has already been added to $q$ in a previous step, or the selected operation $op$ is a conjunction and $l$ is inconsistent with some other literal $l'$ in $q$, taking into account a (possibly empty) set of atemporal domain axioms (see lines 5–6). If none of the above conditions is satisfied, then we add $l$ to the set of literals that will be used to construct $q$ (line 7).

After identifying $l\_no$ literals that are suitable for constraint $\mathsf{A}(\phi)$, we construct $\phi$ and $\mathsf{A}(\phi)$ using the sampled operation $op$ (see lines 8–9 of Algorithm 1). Next, we need to verify whether $\mathsf{A}(\phi)$ is inconsistent or redundant with respect to the constraints that are already present in $C$. To do this, for each constraint $q'$ in $C$, we check if $\mathsf{A}(\phi)$ is compatible with $q'$ (lines 10–17). If $\mathsf{A}(\phi)$ is not compatible with some constraint in $C$, then we drop $\mathsf{A}(\phi)$ and generate another constraint. Otherwise, if $\mathsf{A}(\phi)$ is compatible with every constraint in $C$, then we add it in $C$ (line 18). For example, $q=\mathsf{A}(\phi)$ is compatible with a $\mathsf{Sometime}$ constraint $q'=\mathsf{S}(\phi')$ if, according to the atemporal domain axioms, (i) $\phi$ does not imply $\phi'$, as $\phi \to \phi'$ would imply that constraint $\mathsf{S}(\phi')$ is redundant given $\mathsf{A}(\phi)$; and (ii) $\phi$ and $\phi'$ are consistent, because if they were inconsistent, it would be impossible to satisfy $\mathsf{A}(\phi)$ given that $\mathsf{S}(\phi')$ holds, i.e., in the case where $\phi'$ is true in at least one state of a valid plan.

### A.3 Automated LLM Plan Verifier

---
**Algorithm 2** LLM Plan Verifier

---
**Require:** LLM plan $\pi_{NL}$, compiled problem $\Pi^{cm}$, optimal cost $c^*$
**Ensure:** Plan validation outcome: Invalid, Suboptimal or Optimal
1: $s \leftarrow I$
2: **for** $a_{NL} \in \pi_{NL}$ **do**
3:      **if** $pddl\_format(a_{NL})$ **then** $a \leftarrow extract\_action(a_{NL})$
4:      **else** $a \leftarrow closest\_action\_edit\_distance(a_{NL}, \Pi)$
5:      $s \leftarrow simulate\_action(a, s, \Pi)$
6:      **if** $s=$None **then return** Invalid
7: **if** $s \not\models G$ **then return** Invalid
8: **if** $length(\pi_{NL}) > c^*$ **then return** Suboptimal
9: **return** Optimal

---

Algorithm 2 outlines LEXICON's automated LLM plan verifier. Its input is an LLM-generated plan $\pi_{NL}$, a PDDL problem $\Pi^{cm}$ that has been compiled with `LiftedTCORE` and the optimal cost $c^*$ of $\Pi^{cm}$ that was discovered by LEXICON during constrained problem generation. (Recall that a plan that is valid for the compiled version of a problem satisfies all the constraints in the original problem.) Given this input, Algorithm 2 reports whether $\pi_{NL}$ is an invalid plan, a valid but suboptimal plan, or a valid, optimal plan for problem $\Pi^{cm}$. This is achieved in two steps: (i) simulating plan $\pi_{NL}$ over $\Pi^{cm}$ to verify its validity, and, if $\pi_{NL}$ is valid, (ii) comparing the length of $\pi_{NL}$ to the optimal cost $c^*$ of $\Pi^{cm}$ in order to check whether $\pi_{NL}$ is optimal.

To initiate the simulation of plan $\pi_{NL}$ over problem $\Pi^{cm}$, we set variable $s$, tracking the state of the problem, to the initial state $I$ of $\Pi^{cm}$ (line 1 of Algorithm 2), and iterate over the actions in $\pi_{NL}$, in order to sequentially simulate the effects of each one over $\Pi^{cm}$ (line 2). The prompt we use for LLM plan generation requests a specific format for LLM actions, so that they can be mapped directly to domain actiin in PDDL (line 3). In practice, however, LLM actions may deviate from this format; we handle such cases by mapping the LLM-generated action $a_{NL}$ to the PDDL domain action yielding the shortest edit distance from $a_{NL}$ (line 4). Both cases map $a_{NL}$ to a PDDL domain action $a$, which we apply on the current state $s$ of our plan simulation (line 5). If the application of $a$ does not lead to a new state, then we deduce that either the preconditions of $a$ are not met in state $s$, or that the application of $a$ over state $s$ led to the violation of a constraint of the original problem. Thus, in this

case, we deduce that plan $\pi_{NL}$ is invalid. If the simulation of all LLM-generated actions over $\Pi^{cm}$ succeeds, then we check whether the goal of the problem is satisfied in the final state $s$ reached in the simulation. If the goal is not satisfied in $s$, then plan $\pi_{NL}$ is invalid (line 7). Otherwise, if the goal is satisfied in $s$, then $\pi_{NL}$ is valid, and we proceed with checking whether $\pi_{NL}$ is optimal or not. If the length of $\pi_{NL}$ is greater than $c^*$, i.e., the optimal cost of $\Pi^{cm}$, then plan $\pi_{NL}$ is suboptimal (line 8). Otherwise, the length of $\pi_{NL}$ is equal to $c^*$, and thus $\pi_{NL}$ is an optimal plan for the problem (line 9).

# B  Additional Results

## B.1  Performance of Non-Reasoning LLMs

| Model | Blocksworld | | BabyAI | | Logistics | | Sokoban | | AlfWorld | |
|---|---|---|---|---|---|---|---|---|---|---|
| | Opt. | Val. | Opt. | Val. | Opt. | Val. | Opt. | Val. | Opt. | Val. |
| DeepSeek V3 | 0 | 7% | 9% | 17% | 0 | 5% | 0 | 0 | 15% | 21% |
| Claude 3.7 Sonnet (no thinking) | 0 | 0 | 12% | 20% | 5% | 5% | 0 | 0 | 3% | 12 |
| Gemini 2.0 Pro | 0 | 7% | 0 | 0 | 0 | 0 | 0 | 0 | 0 | 0 |
| OpenAI GPT-4.1 | 3% | 16% | 17% | 26% | 0 | 3% | 0 | 0 | 0 | 0 |

Table 2: Performance of non-reasoning LLMs on problems with one constraint. We measured the percentage of problems solved with an optimal plan (Opt.) and the percentage of problems solved with a valid, but possibly suboptimal, plan (Val.).

We complement the experimental results in Figure 6 of the paper with the performance of LLMs that do not use explicit thinking, i.e., DeepSeek V3, Claude 3.7 Sonnet (no extended thinking) and Gemini 2.0 Pro, on constrained planning problems. Table 2 displays their performance on each domain, in terms of plan optimality and plan validity, on problems that included one constraint. None of these models was able to produce an optimal plan for a problem from our benchmark that included more than one constraint.

## B.2  LLM Action Format Compliance

| Model | Blocksworld | | | BabyAI | | | Logistics | | | Sokoban | | | AlfWorld | | |
|---|---|---|---|---|---|---|---|---|---|---|---|---|---|---|---|
| | 1 | 5 | 10 | 1 | 5 | 10 | 1 | 5 | 10 | 1 | 5 | 10 | 1 | 5 | 10 |
| DeepSeek R1 | 0 | 50% | 47% | 6% | 10% | 23% | 3% | 7% | 10% | 3% | 3% | 3% | 6% | 0 | 0 |
| OpenAI o3 | 0 | 0 | 0 | 0 | 3% | 3% | 0 | 0 | 3% | 0 | 0 | 3% | 0 | 0 | 3% |
| Gemini 2.5 Pro | 0 | 0 | 0 | 0 | 0 | 0 | 0 | 3% | 3% | 0 | 0 | 0 | 70% | 78% | 81% |
| Claude 3.7 Sonnet (with thinking) | 27% | 33% | 47% | 10% | 12% | 27% | 33% | 42% | 60% | 36% | 45% | 96% | 15% | 33% | 39% |
| GPT-5 | 3% | 3% | 3% | 0 | 0 | 0 | 3% | 0 | 0 | 0 | 0 | 0 | 0 | 3% | 0 |

Table 3: Percentage of LLM-generated plans that could not be mapped directly into PDDL for problems with 1, 5 and 10 constraints.

During LLM plan verification, we measured the number of times an LLM-generated plan did not comply with the format we instructed the LLMs to follow via our prompt. Table 3 displays our results on LLMs with reasoning capabilities, when instructed to suggest plans for problems with 1, 5 and 10 constraints. Our results show that the responses of o3 and Gemini 2.5 Pro included, in almost all cases, a plan that conformed with the format of PDDL actions, and could thus be mapped directly into a PDDL plan, without needing to resort to distance calculations between LLM-generated actions and domain actions. In contrast, the responses of R1 and Claude 3.7 Sonnet often deviated from the requested plan format, in which cases we needed to map some of the actions in the suggested plans into PDDL action via distance minimisation, in order to be able to verify these plans.

# C Experimental Setup and Reproducibility

First, we outline the hyperparameter values used in LLM executions. Second, we provide a set of instructions for running LEXICON. Third, we outline the steps for reproducing our experiments.

## C.1 Execution Parameters

| Model | max tokens | temperature |
|---|---|---|
| DeepSeek R1 | 40K | 0.2 |
| OpenAI o3 | 100K | 1 |
| Gemini 2.5 Pro | 64K | 0.2 |
| Claude 3.7 Sonnet (with thinking) | 64K | 1 |
| GPT-5 | 128K | 0.2 |
| GPT-4.1 | 32K | 0.2 |
| DeepSeek V3 | 8K | 0.2 |
| Gemini 2.0 Pro | 8K | 0.2 |
| Claude 3.7 Sonnet (no thinking) | 64K | 0.2 |

Table 4: LLM hyperparameters.

Table 4 displays the values for the upper limit on generated tokens (including both completion and reasoning tokens) and the temperature hyperparameters used for each LLM. For all models, we set the upper limit for generated tokens on the maximum value allowed by their developers. We chose to use a temperature of 0.2, i.e., a low value that enables structured, deterministic thinking, while also being higher than zero, allowing a certain degree of exploration. In the case of o3 and Claude 3.7 Sonnet, we used the temperature value 1, because this was the only temperature value allowed for these models when thinking is enabled.

## C.2 Code Execution Instructions

Our code is publicly available on Github[3]. We provide instructions on executing our constrained planning problem generator and our LLM plan verifier on the domains that are present in our repository. You may add a custom domain by providing a PDDL domain file, an initial state-goal pair generator and NL descriptions of the actions and the atoms of the domain, following the structure of the domains in our repository.

**Installation.** You may install LEXICON by following these steps:

1. Install conda on an Ubuntu machine.

2. Clone our repository with Git.

3. Create a conda environment with the necessary package dependencies installed. To do this, visit the root directory of our repository and run:

    ```
    conda env create ——name lexiconenv ——file=environment.yml
    ```

4. Activate your new conda environment with: `conda activate lexiconenv`

5. Make sure that the following packages are installed: [anthropic==0.51.0, dotmap==1.3.30, gym==0.26.2, gymnasium==1.0.0, hydra-core==1.3.2, matplotlib==3.7.1, minigrid==3.0.0, numpy==2.2.6, omegaconf==2.3.0, openai==1.81.0, protobuf==6.31.0, pyprover==0.6.2, tqdm==4.67.1, unified_planning==1.2.0]

All the instructions that follow require that you have the `lexiconenv` environment activated.

**Constrained Planning Problem Generation.** To generate a constrained planning problem for a specified domain, you may use script `generate_benchmark.py`.

This script receives as input:

---

[3]`https://github.com/periklismant/lexicon_neurips`

- a domain name ("blocksworld", "babyai", "logistics", "sokoban", or "alfworld"),
- an integer denoting the random seed for generating the first problem in the benchmark,
- the number of problems to be generated, and
- the number of constraints in each problem.

The output of the script is:

- a constrained problem for the domain (in both PDDL and NL), located in folder:
  `domains/{domain_name}/data_{constraints_no}/{seed_no}`

In order to run our problem generator, follow these steps:

1. Move into the root directory of our repository.
2. Construct a directory with the name `intermediate_sas`, which is a required folder for SymK to store intermediate computations, with the following command:
   `mkdir intermediate_sas`
3. Run command:
   `python3 generate_benchmark.py {domain_name} {initial_seed} {problems_no} {constraints_no}`

Example executions:

- `python3 generate_benchmark.py blocksworld 100 1 2`
  → Starting from seed 100, construct a blocksworld problem with 2 constraints.
- `python3 generate_benchmark.py logistics 50 3 4`
  → Starting from seed 50, construct 3 logistics problems with 4 constraints each.

**LLM Plan Verification.** To validate an LLM-generated plan for a constrained planning problem, you may use script `verify_plan.py`.

This script receives as input:

- a domain name ("blocksworld", "babyai", "logistics", "sokoban", or "alfworld"),
- a folder number (corresponding to the number of constraints in the generated problem),
- a data number (corresponding to the seed used to generate the problem), and
- an llm name ("deepseek", "o3", "gemini-2.5", "claude_37_sonnet", "gpt_5"), where "deepseek" verifies a plan produced by R1.

The output of the script is:

- an indication on whether the plan stored in
  `domains/{domain_name}/data/data_{folder_no}/{data_no}/{llm}_plan`
  is invalid, suboptimal or optimal.

In order to run our LLM plan verifier, follow these steps:

1. Move into the root directory of our repository.
2. Run command:
   `python3 verify_plan.py {domain_name} {folder_no} {data_no} {llm}`

Example executions (on pre-generated, packed LLM plans):

- `python3 verify_plan.py babyai 1 1 o3`
  → Verifies that the plan in the corresponding directory is optimal.
- `python3 verify_plan.py babyai 3 1 o3`
  → Verifies that the plan in the corresponding directory is invalid.
- `python3 verify_plan.py blocksworld 5 1 o3`
  → Verifies that the plan in the corresponding directory is suboptimal.

## C.3 Experiment Reproducibility Instructions

Reproducing our experiments requires three main steps:

1. Generating benchmarks with problems having an increasing number of constraints for each domain.
2. Evaluating LLMs on the generated benchmarks.
3. Verifying the LLM plans produced in the previous step.

Steps 1 and 3 are described in Appendix C.2.

In order to run LLMs on constrained problems generated by LEXICON, follow these steps:

1. Get API keys by OpenAI, Deepseek, Google and Anthropic, and store them in conda environment variables as follows:

   ```
   conda env config vars set OPENAI_API_KEY=yourkey
   conda env config vars set DEEPSEEK_API_KEY=yourkey
   conda env config vars set GEMINI_API_KEY=yourkey
   conda env config vars set ANTHROPIC_API_KEY=yourkey
   ```

   You have to deactivate and reactivate your conda environment for the variable changes to take effect. In order to use some models, such as o3, you may need to elevate your subscription to a certain tier level.

2. Open file `cfg/config.yaml` with a text editor and make the following changes:
   - set the value of `mode` to `evaluation`.
   - set the value of `folder_no` to the `constraints_no` used to generate the problems you want the LLMs to solve.
   - set the value of list `evaluation_data` to the ids of the problems you want to evaluate LLMs on. These problem ids can be found in:
     `domains/{domain_name}/data/data_{folder_no}/`
   - add a new key-value pair "`llm: evaluation`".

3. Evaluate DeepSeek R1, OpenAI o3, Gemini 2.5 Pro, Claude 3.7 Sonnet (with extended thinking) and GPT-5 on the problems selected in the previous step on, e.g., the Blocksworld domain:

   (a) Create a file named `run_blocksworld.py` and add to it the following code:

   ```
   1  from omegaconf import OmegaConf
   2  from domains.blocksworld.blocksworld import main
   3  if __name__ == "__main__":
   4      cfg = OmegaConf.load("cfg/config.yaml")
   5      main(cfg)
   ```

   (b) Run the following command:
   ```
   python3 run_blocksworld.py
   ```

   You may evaluate these LLMs on a different domain by replacing "blocksworld" with the name of another domain in the above steps.

4. In order to use different LLMs, open file `lexicon.py` with a text editor, make the following changes and then go back to the previous step.
   - Go to the definition of `evaluate_llms` and adjust the elements of list `model_names_and_strategies`.

