# OpenReview forum: "LexiCon: a Benchmark for Planning under Temporal Constraints in Natural Language"
_NeurIPS.cc/2025/Datasets_and_Benchmarks_Track — NeurIPS 2025 Datasets and Benchmarks Track poster_

### Official Review · Reviewer_UgB9 · 2025-06-03

**Rating:** 4
**Confidence:** 3

**Summary:**

This paper introduces LEXICON, an extensible natural language-based benchmark for evaluating Large Language Models (LLMs) on planning tasks with temporal constraints. LEXICON automatically generates constrained planning problems from existing environments by imposing temporal constraints on states, translates these to natural language, and uses a symbolic engine to verify LLM-generated plans for correctness and optimality. Experimental results show that the performance of state-of-the-art LLMs, including reasoning models, significantly decreases as the complexity of constraints increases. The benchmark is designed to be extensible to new environments and increasing difficulty levels.

**Dataset Code Accessibility:**

Yes

**Ethical Considerations:**

No, there are no or only very minor ethics concerns

**Limitations Weaknesses:**

1. While the translation of instance-specific elements (initial state, goal, constraints) is automated via templates for atoms, the domain-level descriptions (e.g., environment and action semantics) are carefully handcrafted per domain. This means adding a completely new domain still requires expert effort for these higher-level NL descriptions.

2. The system relies on PDDL3.0 for its symbolic reasoning engine and constraint specification. While powerful, this means the "natural language" interaction primarily serves as an interface layer for LLMs, with the core logic remaining formal. The complexity of NL understanding is thus tied to how well LLMs can map the provided NL to these underlying formalisms.

3. The benchmark currently supports four environments (BabyAI, Blocksworld, Logistics, Sokoban). While extensible, the initial evaluation relies on this set, which, though varied, might not cover all nuances of constrained planning.

**Strengths Contributions:**

1. LEXICON addresses a critical gap in LLM evaluation by focusing on planning under temporal constraints specified in natural language. This is highly relevant as LLMs are increasingly considered for real-world applications where adherence to constraints (especially safety-related) is paramount.

2. LEXICON incorporates a symbolic reasoning engine for automatically verifying LLM-generated plans for both correctness (goal achievement and constraint satisfaction) and optimality (minimum plan length). This is more reliable than manual or LLM-based verification. It uses tools like SymK for planning and LiftedTCORE for compiling constraints away.

3. The experiments demonstrate a clear trend: LLM performance, even for reasoning models, dramatically deteriorates as the number of constraints increases across various domains (Blocksworld, BabyAI, Logistics, Sokoban). This highlights current LLM limitations in complex constrained planning.

4. The paper includes a qualitative analysis of reasoning traces from R1, identifying behaviors like bilevel planning, backtracking, lookahead search, and self-reflection, while also noting the lack of structured search necessary for guaranteed optimal planning.

---

> ### Author Rebuttal · Authors · 2025-07-31
>
> We would like to thank the reviewer for their valuable review. Below, we provide detailed responses to the points raised by the reviewer.
>
> **Weakness 1**: While the translation of instance-specific elements (initial state, goal, constraints) is automated via templates for atoms, the domain-level descriptions (e.g., environment and action semantics) are carefully handcrafted per domain. This means adding a completely new domain still requires expert effort for these higher-level NL descriptions.
>
> **Response**: Though it is necessary to provide NL descriptions for the environment and the actions in each new domain, these descriptions are usually straightforward to construct in practice. The environment is expressed with a short, high-level description that is independent from the PDDL domain file. The actions are first described based on their high-level usage. Then, the preconditions and the effects of each action are expressed in NL by composing the NL descriptions of the atoms they comprise. **To ease the extension of our benchmark, we will update the code of LexiCon to automate the NL translation of action preconditions and effects**. To do this, we will use recursion to automatically compose the NL templates corresponding to the atoms appearing in action preconditions and effects, using an algorithm that is similar to the one we use to translate the instance-specific elements of a problem. We will make this update to the code within August. Thank you for the feedback!
>
> **Weakness 2**: The system relies on PDDL3.0 for its symbolic reasoning engine and constraint specification. While powerful, this means the "natural language" interaction primarily serves as an interface layer for LLMs, with the core logic remaining formal. The complexity of NL understanding is thus tied to how well LLMs can map the provided NL to these underlying formalisms.
>
> **Response**: We focus on PDDL3.0 because it captures useful temporal constraints for real-world planning, such as safety constraints, while real-world (safety) constrained planning is a task for which LLMs are increasingly being deployed (see lines 36–38). We clarify that, in order **for an LLM to provide an optimal plan for an NL constrained planning problem sample, it does not suffice to translate the NL back to PDDL, as the LLM would additionally need to mimic the execution of a constrained planning algorithm in order to get an optimal plan**. Therefore, though it may be true that NL problem understanding could be achieved by mapping it back to PDDL, it is not straightforward that LLMs operate in this way to solve a problem, and, in any case, **seem to fail at reliably simulating the execution of a constrained planning algorithm based on our experimental results (see Figure 5)**.
>
> Moreover, depending on the expressiveness of the constrained planning problem, **after translating its NL description to PDDL with an LLM, it may be impossible to use an off-the-shelf planner on the resulting PDDL file, as planning engines differ widely on their expressivity**. For instance, FF,  FastDownward and SymK don't work on PDDL 3.0 constraints, while Optic does not capture several constructs that appear in the domains we used, such as conditional effects and negative preconditions in actions. Therefore, we cannot assume that all real-world problems can be solved by asking the LLMs to translate the problem to PDDL and deploying a single planner.
>
> **Weakness 3**: The benchmark currently supports four environments (BabyAI, Blocksworld, Logistics, Sokoban). While extensible, the initial evaluation relies on this set, which, though varied, might not cover all nuances of constrained planning.
>
> **Response**: We employed **an initial set of environments that allows for various types of constraints**. For instance, the constraints in Blocksworld may require or forbid specific configurations of blocks to come about, while the constraints in our 2D gridworld domains (BabyAI and Sokoban) may impose safety requirements in agent navigation, like prohibiting the agent from stepping on specified tiles or visiting a specified room. Moreover, **since the time of submission, we have extended LexiCon with a new planning domain (called ALFWorld) for executing household tasks**, which is used in [1]. The tasks in ALFWorld go beyond navigation and object displacement, as they may additionally dictate that an object must be cleaned, heated, cooled, observed under a light or placed within a receptacle. These tasks may be incorporated in constraints for ALFWorld problems, leading to NL-based planning problems with a varied set of constraints. For instance, the constraint “SometimeBefore(inReceptacle(cup1, cabinet1), isClean(cup1)) expresses that cup1 needs to be cleaned before being placed in kitchen cabinet1. **The household tasks in ALFWorld reflect real-world scenarios, aligning with our motivation of evaluating LLMs in embodied robotic tasks generated by LexiCon in the future**. We will introduce the ALFWorld domain in the LexiCon repository within August.
>
> [1] Mohit Shridhar, Xingdi (Eric) Yuan, Marc-Alexandre Côté, Yonatan Bisk, Adam Trischler, Matthew Hausknecht. Aligning Text and Embodied Environments for Interactive Learning. ICLR 2021.

---

### Official Review · Reviewer_XTdQ · 2025-06-29

**Rating:** 5
**Confidence:** 2

**Summary:**

This paper introduces LexiCon, which is a benchmark for planning under temporal constraints using natural language. It can automatically generate various temporal constraints on PDDL domains, convert them into prompts, and verify LLM outputs by parsing the answer back into PDDL and checking validity and optimality by some PDDL means. The paper tests the method on four domains (BabyAI, Blocksworld, Logistics, Sokoban), and uses the results to show that performance drops a lot as constraints increase. LexiCon’s pipeline enables controlled difficulty scaling and end-to-end automation for realistic planning tasks. The benchmark extends previous unconstrained planning evaluations by embedding temporal constraints expressed in natural language, lowering the barrier for specifying complex requirements.

**Additional Feedback:**

In the current version of the paper, only discrete temporal constraints are included. In the future, the paper may consider adding support of continuous states, probabilistic elements, or partially observable scenarios.

**Dataset Code Accessibility:**

Yes

**Dataset Code Comments:**

The GitHub repository is well organized and provides detailed instructions for running the code. It would benefit from additional usage examples with sample outputs for users to reference, as well as documentation of common errors and troubleshooting tips.

**Ethical Comments:**

I do not find any ethical concern for this paper.

**Ethical Considerations:**

No, there are no or only very minor ethics concerns

**Limitations Weaknesses:**

Please find the limitations below:
1. The benchmark primarily uses accuracy as its metric. Assessing crucial practical factors such as generation time, resource consumption, or robustness can make the benchmark more reliable and easier to use.
2. The verification based on SymK and TCORE has high computational overhead in large-scale and real-time scenarios. I am not sure whether some randomized or heuristic method can be used to optimize the average performance.
3. In real LLM usage, complex or varied natural-language instructions may lead to ambiguity or mapping errors while the benchmark relies on fixed templates. Some randomizations or enumerations of the form of instruction may be useful in real-world tests.

**Strengths Contributions:**

Please find the strengths below:
1. The paper first provides a natural-language-based benchmark for planning within temporal limitations in any PDDL domains. It includes an automated pipeline that integrates cost-aware constraint generation with NL-PDDL translation and verification.
2. The paper includes evaluation across domains (BabyAI、Blocksworld、Logistics、Sokoban) and LLMs, showing performance degradation when constraints are applied, which respects the natural conjecture.
3. The paper provides clear and reproducible writing, and diagrams, tables, and pseudocode are easy to understand to me.

---

> ### Author Rebuttal · Authors · 2025-07-31
>
> We would like to thank the reviewer for their valuable review. Below, we provide detailed responses to the points raised by the reviewer.
>
> **Weakness 1**: The benchmark primarily uses accuracy as its metric. Assessing crucial practical factors such as generation time, resource consumption, or robustness can make the benchmark more reliable and easier to use.
>
> **Response**: Apart from the accuracy of the LLM-generated plans, **LexiCon also measures the time required for LLMs to provide a solution and the number of output and thinking tokens they used (see Figures 7 and 6, respectively)**. Regarding **the time used by LexiCon to generate, solve, and translate problem samples**, we report this metric in the first (yellow) bar of the bar groups **in Figure 7**.
>
> **Weakness 2**: The verification based on SymK and TCORE has high computational overhead in large-scale and real-time scenarios. I am not sure whether some randomized or heuristic method can be used to optimize the average performance.
>
> **Response**: SymK is a state-of-the-art optimal planner that employs search heuristics, while we employ a lifted variant of TCORE to mitigate computational cost of planning on grounded planning specifications. Though planning is a worst-case computationally expensive task, we observe that **constrained problem instance generation and solution with Lexicon is, in practice, at least one order of magnitude faster than planning with LLMs (see Figure 7)**, making real-time evaluation of these LLMs feasible.
>
> **Weakness 3**: In real LLM usage, complex or varied natural-language instructions may lead to ambiguity or mapping errors while the benchmark relies on fixed templates. Some randomizations or enumerations of the form of instruction may be useful in real-world tests.
>
> **Response**: Though we did not observe such mapping errors in the domains we used, we acknowledge that, as more complicated planning domains are added to the benchmark, such errors may come about. **We will allow support for multiple variants of the NL descriptions for domain atoms and actions, as well as of the ones for logical and constraint operators, aiming to alleviate possible ambiguities**. For example, our set of NL descriptions expressing constraint “Always(x)” will include, among others, the following templates.
>
> - “Expression x must always be true”.
>
> - “Expression x must hold in every state”
>
> - “Expression x must remain true throughout the execution of a plan”
>
> Each time an Always constraint needs to be expressed in NL, our translator will randomly choose a subset of our NL templates for “Always(x)” and communicate them to the LLM.
>
> We expect to finish this update in the code within August.
>
> **Dataset Code Comments**: The GitHub repository is well organized and provides detailed instructions for running the code. It would benefit from additional usage examples with sample outputs for users to reference, as well as documentation of common errors and troubleshooting tips.
>
> **Response**: We will add such examples and update the documentation accordingly. Thank you for the feedback!
>
> **Additional Feedback**: In the current version of the paper, only discrete temporal constraints are included. In the future, the paper may consider adding support of continuous states, probabilistic elements, or partially observable scenarios.
>
> **Response**: Thank you for the suggestion. **We are indeed considering these extensions as possible future work directions, as they may be beneficial for evaluating reinforcement learning agents on LexiCon, which is another goal of ours**. We will start by considering actions with non-deterministic effects, as they may be used to model environment reactions to agent actions, which are often present in reinforcement learning tasks. Some extensions of PDDL that encapsulate such actions are Fully-Observable Non-Deterministic planning and Probabilistic PDDL. Both extensions complicate the planning task, and would thus require constrained planning and verification techniques that go beyond the current capabilities of LexiCon. We plan to investigate such techniques in the near future.

---

### Official Review · Reviewer_JeMb · 2025-07-02

**Rating:** 5
**Confidence:** 3

**Summary:**

This paper introduces LEXICON, an extensible NL-based benchmark for planning with temporal constraints specified on state-trajectories. They have evaluated several state-of-the-art LLMs on benchmarks generated by LEXICON. The results indicate that LLM performance consistently declines with the number of constraints, suggesting that current models do not yet apprehend and apply formal planning algorithms.

**Additional Feedback:**

Based on the insights from this study, what approaches do the authors recommend to improve an LLM’s performance on the benchmarks generated by LEXICON?

**Dataset Code Accessibility:**

Yes

**Dataset Code Comments:**

The authors provide the data and the code we used in our submission, and reproducibility instructions in Appendix C. The authors also provide the LLM parameters in Appendix C.1

**Ethical Considerations:**

No, there are no or only very minor ethics concerns

**Final Justification:**

The authors have addressed my concerns thoroughly and clarified the difference between their approach and some related work during the rebuttal. The proposed method is both promising and important for designing evaluations of LLM robustness. Therefore, I recommend this paper for acceptance.

**Limitations Weaknesses:**

1. The discussion in the related work section is limited. There are additional related works worth discussing, such as [1–3].

[1] Ding, Zifeng, et al. "TCP: a Benchmark for Temporal Constraint-Based Planning." arXiv preprint arXiv:2505.19927 (2025).

[2] Aghzal, Mohamed, Erion Plaku, and Ziyu Yao. "Can large language models be good path planners? a benchmark and investigation on spatial-temporal reasoning." arXiv preprint arXiv:2310.03249 (2023).

[3] Wang, Yuqing, and Yun Zhao. "Tram: Benchmarking temporal reasoning for large language models." arXiv preprint arXiv:2310.00835 (2023).

2. The concepts, such as Automated Curation, Suite Extensibility and Environment Diversity, are not well defined in the paper.

3. How to ensure that samples generated by LEXICON are both high-quality and sufficiently diverse?

**Strengths Contributions:**

1. The paper is well-written and easy to understand. All figures and tables are informative.

2. This work proposes an extensible NL-based benchmark generator, LEXICON, which is promising to be used to design benchmarks to evaluate the robustness of current LLM models.

3. The evaluation setup is comprehensive, featuring four up-to-date LLMs assessed across four diverse datasets.

---

> ### Author Rebuttal · Authors · 2025-07-31
>
> We would like to thank the reviewer for their valuable review. Below, we provide detailed responses to the points raised by the reviewer.
>
> **Weakness 1**: The discussion in the related work section is limited. There are additional related works worth discussing, such as [1–3].
>
> [1] Ding, Zifeng, et al. "TCP: a Benchmark for Temporal Constraint-Based Planning." arXiv preprint arXiv:2505.19927 (2025).
>
> [2] Aghzal, Mohamed, Erion Plaku, and Ziyu Yao. "Can large language models be good path planners? a benchmark and investigation on spatial-temporal reasoning." arXiv preprint arXiv:2310.03249 (2023).
>
> [3] Wang, Yuqing, and Yun Zhao. "Tram: Benchmarking temporal reasoning for large language models." arXiv preprint arXiv:2310.00835 (2023).
>
> **Response**: Thank you for suggesting these related papers. Below, we contrast the benchmarks provided in these papers with LexiCon, commenting on the comparison axes in Table 1 when useful (see our response to “Weakness 2” for a more rigorous formulation of these comparison axes).
>
> The TCP benchmark [1] produces planning problems with temporal constraints by observing agent dialogues on collaborative projects. In TCP, the goal of the LLM being evaluated is to infer an optimal schedule of the subtasks required for the project that satisfies all temporal constraints. Contrary to LexiCon, **TCP lacks environment diversity as all domains considered are variants of project discussions that only differ in the field of study** (e.g., Mathematics, Biology, etc.). Moreover, the problem considered in **TCP is more akin to scheduling and question answering** — as it tasks the LLM with answering a project-related question given temporal constraints that restrict the working hours of project group members — and not to planning, in the sense of identifying a sequence of actions that solve a problem, which is the task being assessed in LexiCon. While scheduling is an NP-hard problem [i], planning is in PSPACE [ii], and thus **the planning problems considered in LexiCon are computationally harder than those in TCP in the worst-case**.
>
> The PPNL benchmark [2] comprises 2D gridworld navigation problems where an agent is tasked to reach one or multiple goal locations, while avoiding objects and adhering to ordering constraints with respect to goal satisfaction. Contrary to LexiCon, **the PPNL benchmark is not extensible and lacks domain variety, as GridWorld is the only domain considered**. Moreover, **the class of constraints considered in PPNL (goal ordering constraints) lacks variety and compositionality** compared to the ones supported in LexiCon. Our benchmark additionally expresses, inter alia, safety constraints, and allows arbitrary combinations of domain atoms in a constraint formula via logical operators, leading to more involved constraints.
>
> The TRAM benchmark [3] assesses LLMs in various temporal reasoning tasks, such as temporal causality and event frequency. Contrary to LexiCon, the tasks included in TRAM constitute question-answering problems, and not planning problems. Moreover, **the question-answering problems in TRAM require fewer reasoning steps than the constrained planning problems generated by LexiCon**.
>
> We will update the related work section with the above discussion.
>
> [i] Garey, M. R., & Johnson, D. S. “Computers and intractability: A guide to the theory of NP-completeness.” W.H. Freeman, 1979.
>
> [ii] Bylander T. "The computational complexity of propositional STRIPS planning”, Artificial Intelligence, 69(1–2), 165–204.
>
> **Weakness 2**: The concepts, such as Automated Curation, Suite Extensibility and Environment Diversity, are not well defined in the paper.
>
> **Response**: In our literature review, we use these concepts as having the following meaning.
>
> *Automated Curation*: The ability of a benchmark to automatically generate new planning problem instances and verify solutions for those instances.
>
> *Suite Extensibility*: The support for extending a benchmark with new planning domains without needing to rewrite its code.
>
> *Environment Diversity*: A benchmark fulfills the environment diversity requirement if it supports more than one planning domain.
>
> We will add the above formulations of these concepts in the next version of our paper.
>
> **Weakness 3**: How to ensure that samples generated by LEXICON are both high-quality and sufficiently diverse?
>
> **Response**: **Regarding the quality of the samples**, given an unconstrained planning problem, **our generator produces constraints that are task-aware**, in the sense that they complicate the original (unconstrained) problem, thus leading to a constrained planning problem that is meaningful for agent evaluation. To do this, we invoke the off-the-shelf optimal planner SymK, over both the unconstrained problem and its version including the generated constraints, and compare the optimal plans of the two problem versions. If the optimal plan for the constrained problem is longer, then we deem that the problem sample is meaningful for our benchmark. Moreover, before adding a new constraint to the problem, **our generator assesses whether the new constraint is consistent with, it does not imply and it is not implied by any one of the previously sampled constraints**, thus leading to a planning problem with a meaningful set of constraints.
>
> **Regarding problem sample diversity**, for each planning domain, both the unconstrained problem instance generation and the constraint generation steps **leverage pseudorandom number generators to diversify various elements of the produced sample**, such as the number of objects in the planning problem, the type of constraints used (e.g., Always, Sometime, etc.), and the number of literals in each constraint. Moreover, **our constraint generator may consider arbitrary combinations of atoms for inclusion in a constraint**. The possible combinations of these atoms increases exponentially with the number of objects in the problem. This plethora of possible formulas that may be included in constraints enables LexiCon to generate highly diverse problem samples.
>
> **Additional Feedback**: Based on the insights from this study, what approaches do the authors recommend to improve an LLM’s performance on the benchmarks generated by LEXICON?
>
> **Response**: While outside the scope of this paper, this is indeed a stimulating question. We see two principled approaches:
>
> - Integrating an LLM with a formal planner in a framework where **the LLM translates the domain and problem files to PDDL, while the planner to solve the resulting PDDL planning task** (cf., LLM+P: Empowering Large Language Models with Optimal Planning Proficiency, Liu et al., 2023).
>
> - Using LLM-Modulo frameworks, wherein, given the NL specification, **the LLM interacts with a planner and gets feedback** — e.g., on constraint satisfaction/violation verdicts — **that enables the LLM to revise its proposed solutions**, forbidding the ones that violate constraints (cf., LLMs Can't Plan, But Can Help Planning in LLM-Modulo Frameworks, Kambhampati et al., 2024).
>
> We will add the above discussion in a dedicated section in the Appendix for the interested reader.

---

> > ### Comment · Reviewer_JeMb · 2025-08-07
> >
> > Thank you for the response. All of my concerns have been addressed, and I will keep my positive score.

---

### Official Review · Reviewer_7GG3 · 2025-07-03

**Rating:** 5
**Confidence:** 3

**Summary:**

This paper introduces LexiCon, a benchmark to evaluate LLMs on planning tasks including temporal constraints expressed in natural language. LexiCon works by augmenting standard planning environments with automatically generated temporal constraints and translating them into natural language prompts. The benchmark is extensible, allowing new environments and increasingly difficult tasks as LLM capabilities advance. Evaluations on several reasoning LLMs show that even SOTA reasoning models struggle as constraints grow more complex.

**Additional Feedback:**

1. Lines 46-47: Does it also guarantee that if a constraint is subsumed by a combination of 2 or more constraints but not subsumed by any one of them individually, it is still considered redundant?

2. This paper probably requires high familiarity with the PDDL 3.0.
I was really confused about where the constraints are coming from in Figure 3 and Sec. 3.2. Its when I looked at the supplementary PDF, I realised they are probably part of the domain file (based on Section A.1). Then I looked at the Kaggle dataset provided with the submission, and found constraints in the problem.pddl files. But according to Figure 1, the problems are autonomously generated (sampled), so I am really confused as to where the initial set of constraints are defined and who provides them. This is really not clear from the main paper, and I strongly suggest restructuring the paper to include the necessary information about what a domain and problem file contains in the main paper.

3. Lines 113-116: When a domain file already supports constraints, what’s the need to have a separate interface for atemporal constraints? Can’t they be added directly to the domain file?

4. Is there any way to verify correctness of the NL translation. In the recent times, there are many papers which show that LLMs are not really great at translating formal specifications to natural language, so how do you handle errors in that part of the work?

**Dataset Code Accessibility:**

Yes

**Dataset Code Comments:**

I was able to access the dataset and see the domain and problem files, both constrained and unconstrained. I was also able to access the code to generate constraints autonomously. The only area of improvement would be to properly document the code, especially the functions in the GitHub repo provided.

**Ethical Considerations:**

No, there are no or only very minor ethics concerns

**Final Justification:**

I have updated my score post author response. I think the paper does a very good job of creating a new benchmark for symbolic planning with temporal constraints for LLMs.

**Limitations Weaknesses:**

1. The paper is confusing about where constraints originate. Figure 3 shows constraints as output from the generator, but the text and examples suggest they might come from domain files. This fundamental aspect needs clearer explanation.

2. The paper assumes deep familiarity with PDDL 3.0, making it less accessible to the broader ML/NLP community who might benefit from understanding LLM planning limitations.

**Strengths Contributions:**

1. LexiCon generates task-aware constraints that complicate planning problems while guaranteeing solvability - a non-trivial novel contribution.

2. New planning domains can be easily added by providing just a PDDL domain file and state-goal generator, making the benchmark future-proof.

---

> ### Author Rebuttal · Authors · 2025-07-31
>
> We would like to thank the reviewer for their valuable review and for taking the time to access our dataset and use our constraint generator. Below, we provide detailed responses to the points raised by the reviewer.
>
> **Weakness 1**: The paper is confusing about where constraints originate. Figure 3 shows constraints as output from the generator, but the text and examples suggest they might come from domain files. This fundamental aspect needs clearer explanation.
>
> &
>
> **Additional Feedback 2**: This paper probably requires high familiarity with the PDDL 3.0. I was really confused about where the constraints are coming from in Figure 3 and Sec. 3.2. Its when I looked at the supplementary PDF, I realised they are probably part of the domain file (based on Section A.1). Then I looked at the Kaggle dataset provided with the submission, and found constraints in the problem.pddl files. But according to Figure 1, the problems are autonomously generated (sampled), so I am really confused as to where the initial set of constraints are defined and who provides them. This is really not clear from the main paper, and I strongly suggest restructuring the paper to include the necessary information about what a domain and problem file contains in the main paper.
>
> **Response**: Our generator produces constraints that complicate an unconstrained planning problem (lines 109–111) and **adds these constraints to its problem file** (**not in the domain file**). This is because constraints are instance-specific information, like the initial state and the goal of the problem, which belong to the problem file. On the other hand, domain files include instance-agnostic information, like the possible actions of the agent, that remain the same for all possible problem instances.
>
> To avoid possible confusion, we clarify that the “domain axioms” shown in Figure 3 are only used to guide constraint generation towards constraints that complicate the original problem. While being provided by humans, domain axioms are not temporal constraints, they are not used during planning, they are not included in neither the domain nor the problem file, and they are completely optional.
>
> Also, we clarify that there is no initial set of constraints; our generator constructs constraints by first composing domain atoms using Boolean operators into a propositional formula, and subsequently placing the resulting formula in an argument of a constraint operator (e.g., Always, Sometime, etc.).
>
> In the case of acceptance, we will restructure the paper accordingly and use part of the extra content page to clarify the structure of the domain and the problem files, and the constraint generation procedure.
>
> Please refer to our response to “Weakness 2” for our answer to the point about the PDDL3.0 familiarity requirement.
>
> **Weakness 2**: The paper assumes deep familiarity with PDDL 3.0, making it less accessible to the broader ML/NLP community who might benefit from understanding LLM planning limitations.
>
> **Response**: We clarify that we did not add a lengthy overview of PDDL3.0 in the main paper because **using and extending our framework with a new domain does not require extensive knowledge of PDDL3.0**. Specifically, introducing a new domain in LexiCon requires a domain file and a problem instance generator (lines 126–127). The domain file employs PDDL constructs (like action operators) that were introduced earlier in PDDL than version 3.0. The only construct from PDDL3.0 we use are the constraints, which are automatically generated by LexiCon, i.e., **the user is never required to write constraints in PDDL3.0 format**.
>
> Moreover, **to aid users that are not familiar with any PDDL variant**, we allow the domain file to be provided within the “Unified Planning” Python library, where **the user may express domain file elements, such as action operators, with (a possibly more friendly) Python syntax** (lines 127–128).
>
> In the case of acceptance, we will use part of the allowed extra content page to provide a more detailed overview of PDDL3.0, similar to the one in Appendix A1.
>
> **Dataset Code Comment**: I was able to access the dataset and see the domain and problem files, both constrained and unconstrained. I was also able to access the code to generate constraints autonomously. The only area of improvement would be to properly document the code, especially the functions in the GitHub repo provided.
>
> **Response**: We will improve the documentation for the functionalities of our system and possibly add a user manual in the future. Thank you for the feedback.
>
> **Additional Feedback 1**: Lines 46-47: Does it also guarantee that if a constraint is subsumed by a combination of 2 or more constraints but not subsumed by any one of them individually, it is still considered redundant?
>
> **Response**: We perform this check with each individual previously sampled constraint, and not over the entire set of previously sampled constraints, because it is a computationally expensive step. When a new constraint is sampled by our generator, we check whether it implies, is implied by or is incompatible with—given an optional set of first-order logic domain axioms—each constraint already added to the problem by our generator in an earlier step. This reduces to a first-order logic theorem-proving task, which can be computationally intensive.
>
> **Additional Feedback 3**: Lines 113-116: When a domain file already supports constraints, what’s the need to have a separate interface for atemporal constraints? Can’t they be added directly to the domain file?
>
> **Response**: These **atemporal constraints (domain axioms) are completely optional and serve as inductive biases to guide the generator toward producing meaningful temporal constraints**. They primarily act as consistency checks (e.g., ensuring only one block can be on top of another in Blocksworld or that an agent cannot be in multiple locations simultaneously in BabyAI), helping to cut down the search space of constraints (by dropping unsatisfiable constraints right away) and making the simulator faster. Note, even in its absence, the Constrained Planning Problem Generator works fines, just a tad bit slower.
>
> **Additional Feedback 4**: Is there any way to verify correctness of the NL translation. In the recent times, there are many papers which show that LLMs are not really great at translating formal specifications to natural language, so how do you handle errors in that part of the work?
>
> **Response**: We clarify that we do **not** use LLMs to translate the PDDL specifications of a planning problem into NL. Instead, **we employ handcrafted NL templates for domain atom descriptions and composition structures** (such as Boolean and constraint operators) **to construct the NL representation of a planning problem compositionally** (see lines 132–134), **ensuring the correctness of the translation with respect to these NL templates**. For instance, see Example 3 of the paper, where the goal “∃v: typeof(v, ball)∧objectColor(v, red)∧at(v)” is translated into NL by composing the NL descriptions of “typeof”, “objectColor” and “at” via NL templates expressing the existential quantifier and the conjunction operator. As a result of following this method, the NL translation is deterministic in the sense that, given the same NL templates for atoms and operators, our translator always provides the same NL problem representation. Thus, correcting translation errors amounts to revising the provided NL templates.

---

> ### Comment · Reviewer_7GG3 · 2025-08-01
>
> Thank you for the detailed response. I see that I misunderstood a few components (like where constraints are defined, and that NL translation is template based). I strongly suggest adding the relevant details in the paper, if accepted, as some of these details were not really explicit in the paper. I am updating the score to "Accept" based on the response.

---

### Decision · Program_Chairs · 2025-09-18

**Decision:**

Accept (poster)

**Comment:**

This paper introduces LexiCon, a novel and extensible benchmark designed to evaluate the planning capabilities of Large Language Models (LLMs) under temporal constraints expressed in natural language. The core contribution is a complete pipeline that takes existing planning domains (specified in PDDL), automatically generates meaningful and solvable temporal constraints, translates the entire constrained problem into a natural language prompt, and then uses a formal verification engine to check the correctness and optimality of the LLM's generated plan.
The authors conduct experiments on four diverse planning domains (BabyAI, Blocksworld, Logistics, and Sokoban) using several state-of-the-art LLMs, including specialized reasoning models. A key finding is that the performance of all tested LLMs degrades significantly as the number and complexity of constraints increase, highlighting a critical limitation in their current reasoning and planning abilities. The benchmark's design is extensible, allowing the community to easily add new domains and scale the difficulty of tasks as LLM capabilities evolve.

Overall reviewers found It addresses a critical gap in LLM evaluation by focusing on constrained planning, which is essential for real-world deployment. The experiments provide clear evidence of the limitations of current LLMs in this domain.
There were concerns around accessibility, scope/definitions and evaluation/robustness. The authors provided good justification during the rebuttal, with inclusion of additional related work, clarification around constrained generation, and the possibility of automatic extension to new domains.